# Effects of Fresh Watermelon Consumption on the Acute Satiety Response and Cardiometabolic Risk Factors in Overweight and Obese Adults

**DOI:** 10.3390/nu11030595

**Published:** 2019-03-12

**Authors:** Tiffany Lum, Megan Connolly, Amanda Marx, Joshua Beidler, Shirin Hooshmand, Mark Kern, Changqi Liu, Mee Young Hong

**Affiliations:** School of Exercise and Nutritional Sciences, San Diego State University, San Diego, CA 92182, USA; tiffanynlum@gmail.com (T.L.); megcconnolly@gmail.com (M.C.); amandamarx27@gmail.com (A.M.); beidler@gmail.com (J.B.); shooshmand@sdsu.edu (S.H.); kern@sdsu.edu (M.K.); changqi.liu@sdsu.edu (C.L.)

**Keywords:** watermelon, satiety, oxidative stress, antioxidant, human

## Abstract

Although some studies have demonstrated the beneficial effects of watermelon supplementation on metabolic diseases, no study has explored the potential mechanism by which watermelon consumption improves body weight management. The objective of this study was to evaluate the effects of fresh watermelon consumption on satiety, postprandial glucose and insulin response, and adiposity and body weight change after 4 weeks of intervention in overweight and obese adults. In a crossover design, 33 overweight or obese subjects consumed watermelon (2 cups) or isocaloric low-fat cookies daily for 4 weeks. Relative to cookies, watermelon elicited more (*p* < 0.05) robust satiety responses (lower hunger, prospective food consumption and desire to eat and greater fullness). Watermelon consumption significantly decreased body weight, body mass index (BMI), systolic blood pressure and waist-to-hip ratio (*p* ≤ 0.05). Cookie consumption significantly increased blood pressure and body fat (*p* < 0.05). Oxidative stress was lower at four week of watermelon intervention compared to cookie intervention (*p* = 0.034). Total antioxidant capacity increased with watermelon consumption (*p* = 0.003) in blood. This study shows that reductions in body weight, body mass index (BMI), and blood pressure can be achieved through daily consumption of watermelon, which also improves some factors associated with overweight and obesity (clinicaltrials.gov, NCT03380221).

## 1. Introduction

Obesity, which affects 39.8% of U.S. adults, contributes to numerous health problems, including cardiovascular disease, hypertension, type II diabetes, and other leading causes of mortality [1,2]. Common treatments for obesity include medications and diets that restrict total calories or specific macronutrients. However, obesity medications are associated with adverse effects, and only about one-fifth of dieters maintain their weight loss for at least one year [3,4]. In light of the serious health consequences of obesity and the limited success of current therapies, there is an urgent need for new approaches.

Healthful snacking may be a simpler and more sustainable approach to improving dietary quality, increasing satiety, and combating weight gain. Over a 40-year period, self-reported eating behaviors of adults demonstrated increased energy intake when snacks were consumed between lunch and dinner or eaten in place of meals [5]. Therefore, choosing snacks that are lower in calories could contribute to a significant reduction in total energy intake. As a snack, fruit is palatable and convenient, and has generally been associated with lower body weight in epidemiological studies [6]. The tendency of fruit to promote a healthy body weight may result from its high water and fiber content, which results in a lower energy density compared with many other popular snack foods [7]. Because people tend to eat a consistent weight of food during a meal or snack, selecting foods that are lower in energy density can lead to greater satiety and lower total energy intake [7].

Fruit has been associated with greater satiety [8,9,10] and lower subsequent energy intake [9,10,11] compared with refined carbohydrate snacks. Although the effects of watermelon on satiety and body weight have not been investigated, watermelon is a strong candidate to promote satiety because its high water content results in a lower energy density than most fruits. One cup of diced watermelon contains only 46 kilocalories but meets 21% of the daily requirement for vitamin C and 17% of the daily requirement for vitamin A [12]. In addition to aiding in weight management, fruit snacks can improve diet quality [13]. Replacing conventional snack foods with watermelon could increase intake of potassium and dietary fiber, which are under-consumed, while reducing intake of added sugars and saturated and *trans* fats, which are consumed in excess [14]. Watermelon has been described as a functional food due to its possible health benefits [15]. Red-fleshed watermelon varieties are rich in lycopene, a carotenoid that may protect against cancer and cardiovascular disease [16]. Watermelon is also the richest food source of l-citrulline, a non-essential amino acid that functions as a precursor for nitric oxide synthesis [17]. Watermelon consumption has been linked to lower blood pressure in humans [17] and improved blood lipid profile in animals and humans [18,19,20,21].

The purpose of this study was to compare the effects of watermelon and an isocaloric low-fat cookie snack on body weight, appetitive sensations, postprandial glucose and insulin, and appetite-regulating hormone concentrations. To determine the effects of chronic watermelon consumption, physiological and metabolic outcomes were measured before and after four weeks of the two snacks. Additionally, acute effects were determined by measuring perceived appetite sensations and blood concentrations of glucose, insulin, and appetite-regulating hormones before and up to 120 min after consumption of the two snacks. We hypothesized that watermelon consumption would reduce body weight by increasing perceived satiety and moderating postprandial glucose and insulin responses compared with an isocaloric low-fat cookie snack.

## 2. Materials and Methods

### 2.1. Participants

Overweight and obese adults (males *n* = 20, females *n* = 13) between the ages of 18 and 55 years with a body mass index (BMI) of 25–40 kg/m^2^ were recruited in Southern California. Exclusion criteria included pregnancy, smoking, any medical problems or metabolic disorders that might alter appetite or body weight, and allergies to or dislike of watermelon (WM) or any ingredient in low-fat cookies (LFC) such as gluten, milk protein, and eggs. Individuals who were actively dieting or engaged in weight loss activities were also excluded, as were women with irregular menstrual cycles. The study was approved by the San Diego State University Institutional Review Board. Potential participants were screened for eligibility criteria, and informed written consent was obtained (clinicaltrials.gov, NCT03380221).

### 2.2. Study Design

This study utilized a crossover design with two 4-week dietary interventions separated by a 2–4 week washout period to prevent carryover effects. Based on a previous human trial of watermelon [22], power analysis (G*Power, Germany) indicated that significant differences would be detected with a sample of 33 subjects at 75% power and an alpha-level of *p* < 0.05. Eligible participants (*n* = 33) were assigned to a 4-week repeated measures crossover with two treatments given—a WM snack followed by a 2–4 week washout period, and then crossed over to an isocaloric-matched LFC snack. Subjects visited the lab at baseline and after 4 weeks for each intervention. Female participants started each trial at day 3 to 11 of their menstrual cycles. Visits occurred in the morning after a 10-h overnight fast. Height and weight (Detecto weigh beam eye-level; ebb City, MO, USA), body fat (dual-energy X-ray absorptiometry, Prodigy, GE Healthcare, Chicago, IL, USA), waist circumference, hip circumference, and blood pressure (Omron M3; Kyoto, Japan) were assessed for each visit. 

During the baseline visits, each subject was instructed on completion of a visual analogue scale (VAS) to assess baseline appetite [23]. Blood samples were collected, then subjects were instructed to consume 2 cups of fresh WM (92 kcal) or isocaloric LFC (92 kcal, Nilla Wafers Reduced Fat, Nabisco, East Hanover, NJ, USA) along with 8 fl. Oz of water in the laboratory. Postprandial responses were measured by administering new VAS to subjects at 20, 40, 60, 90, and 120 min following snack consumption. A second blood sample was collected 60 min post-snack consumption. Blood samples were centrifuged at 1200× *g* for 10 min at 4 °C and serum samples were stored at −80 °C until analysis. 

During the WM intervention, participants consumed 2 cups of fresh diced WM (92 kcal) daily for 4 weeks. During the LFC intervention, participants consumed Nabisco vanilla wafer cookies (92 kcal) daily for 4 weeks. Each WM serving contained 92 kcal, 23 g carbohydrate, 2 g protein, 0 g fat, and 1 g fiber. Each LFC serving contained 92 kcal, 18.2 g carbohydrate, 0.76 g protein, 1.14 g fat, and 0 g fiber. Participants could consume their snacks at any time of day, during one or multiple sittings, alone or in combination with other foods in order to resemble snacking conditions in everyday life. Participants were asked to avoid consuming LFC during the WM intervention and to avoid consuming WM during the LFC intervention, in order to keep the potential effects of the two snacks separate. Aside from the daily consumption of either snack, participants were instructed to maintain their typical dietary intakes and physical activity levels. At the end of each four-week intervention, fasting blood samples were collected.

### 2.3. Satiety Questionnaire

The visual analogue scale (VAS) [23] measured appetite responses by asking a series of 5 questions assessing hunger, fullness, desire to eat, prospective food consumption, and thirst. Each question was followed by a 10 cm line with words anchored at each end, expressing the lowest (0 cm) and highest (10 cm) ratings of each. Subjects could record their responses by marking a spot on the line indicating their feelings about each question. Responses were quantified by measuring the distance from the left end of the line to the designated mark.

### 2.4. Dietary Assessment and Physical Activity

Dietary intake was measured using 24-h dietary recalls on the two days preceding the in-laboratory visit by trained staff. The United States Department of Agriculture (USDA) Supertracker (2012) was used to evaluate average daily energy intakes. Physical activity levels were also assessed using a validated Physical Activity Recall (PAR) Questionnaire [24].

### 2.5. Postprandial Glucose and Insulin Response

Serum glucose was measured using a colorimetric glucose assay kit (Stanbio Laboratory, Boerne, TX, USA) and analyzed according to the manufacturer’s instructions. A reagent containing glucose oxidase was mixed with the serum samples in cuvettes, resulting in the formation of hydrogen peroxide. The hydrogen peroxide then reacted to form a quinone complex. The absorbance of the quinone complex was read at 500 nm. 

Insulin was measured with a sandwich enzyme-linked immunosorbent assay (ELISA) kit (ALPCO, Salem, NH, USA). The serum samples and secondary antibody were added to the insulin-specific antibody coated wells. A 3,3′,5,5′-tetramethylbenzidine (TMB) substrate was added to the wells to produce color and the plates were incubated a second time. Lastly, a stop solution was added and the plates were measured spectrophotometrically at 450 nm.

### 2.6. Appetite-Regulating Hormones

The appetite-regulating hormones leptin, ghrelin, adiponectin, and cholecystokinin (CCK) were assayed using ELISA kits (RayBiotech, Norcross, GA, USA). In each assay, the target hormone in the samples was bound by immobilized antibodies. A biotinylated antibody was added, followed by horseradish peroxidase-conjugated streptavidin, and finally a substrate solution that developed color in proportion to the amount of hormone bound. Following the addition of stop solution, the plates were measured spectrophotometrically at 450 nm.

### 2.7. C-Reactive Protein

C-reactive protein (CRP) was measured using an ELISA kit (Calbiotech, Spring Valley, CA, USA) and analyzed per the manufacturer’s instructions. Microplates were coated with monoclonal antibodies to CRP. The serum samples and anti-CRP-horseradish-peroxidase conjugated secondary antibody were added to the microplate wells. TMB substrate was added to produce color. The plates were incubated, stop solution was added, and absorbance was measured at 450 nm.

### 2.8. Serum Lipids

Blood lipid profiles were analyzed using serum triglyceride (TG), total cholesterol (TC) and high-density lipoprotein cholesterol (HDL-C) (Stanbio Laboratory, Boerne, TX, USA) colorimetric assay kits. Low-density lipoprotein cholesterol (LDL-C) was calculated using the following equation: LDL cholesterol = total cholesterol − HDL cholesterol − (triglycerides/5) [25].

### 2.9. Thiobarbituric Acid Reactive Substances (TBARS)

Lipid peroxidation as a marker of oxidative stress was analyzed using a thiobarbituric acid reactive substances (TBARS) assay kit (Cayman Chemical Company, Ann Arbor, MI, USA). The assay standards were prepared with malondialdehyde. The standards and serum samples were mixed with sodium dodecyl sulfate solution in vials, then the thiobarbituric acid was added. Vials were boiled for 1 h, and then placed in an ice bath. Samples were centrifuged and pipetted into microplate wells. Absorbance was measured at 535 nm.

### 2.10. Catalase Activity

Catalase activity, a marker of antioxidant capacity, was analyzed using a catalase assay kit (Cayman Chemical Company). Serum samples were mixed with hydrogen peroxide to initiate the formation of formaldehyde. Potassium hydroxide was added to stop the reaction and chromogen was added for colorimetric measurement of formaldehyde production. Catalase potassium periodate was added, then absorbance was read at 540 nm.

### 2.11. Total Antioxidant Capacity

Total antioxidant capacity (TAC) was analyzed using an antioxidant assay kit (Cayman Chemical Company). Standards were prepared using Trolox. The standards and serum samples were mixed with metmyoglobin and chromogen and then hydrogen peroxide was added to initiate the reactions. Inhibition of the oxidation of 2,2′-Azino-di-(3-ethylbenzothiazoline sulphonate) (ABTS) by metmyoglobin was measured by reading the absorbance at 405 nm using a spectrophotometer.

### 2.12. Liver Function Markers

Alkaline phosphatase (ALP), alanine aminotransferase (ALT), aspartate aminotransferase (AST), creatine kinase (CK), and lactate dehydrogenase (LDH) were determined in serum samples using assay kits from Stanbio (Boerne, TX, USA). All assays were performed according to the manufacturer’s instructions. ALP was read at 405 nm and the other enzyme assays were read at 340 nm.

### 2.13. Statistical Analysis

The normality of the data was checked using the *explore* function in SPSS (SPSS Statistics version 24, IBM, Armonk, NY, USA). When data showed a normal distribution, two-way repeated measures ANOVA were used to analyze the effects of WM and LFC treatments on each of the variables over time, as well as any significant interactions between snack type and time. If the interaction was significant, a least significant difference (LSD) analysis was used for post-hoc comparison. When normality assumption was suspect, a nonparametric Wilcoxon analysis was performed. Data were considered statistically significant when *p* < 0.05.

## 3. Results

### 3.1. Body Weight and Blood Pressure

Significant interactions were found for body weight (Interaction *p* = 0.005; Snack effect *p* = 0.064; Time effect *p* = 0.730) and BMI (Interaction *p* = 0.006; Snack effect *p* = 0.061; Time effect *p* = 0.799). Body weight and BMI values did not differ significantly between the two interventions at baseline, but were significantly lower after four weeks of WM consumption and significantly higher after four weeks of LFC consumption compared with baseline values (Table 1). Four weeks of WM consumption did not change systolic blood pressure (SBP) and diastolic blood pressure (DBP), but four weeks of LFC consumption significantly increased both SBP (*p* = 0.015) and DBP (*p* = 0.001). Waist-to-hip ratio was lowered at week four of WM consumption compared with week for of LFC consumption (*p* = 0.019). Body fat percentage did not change significantly over the course of the WM intervention but was significantly higher at week four of the LFC intervention than at baseline (*p* = 0.005). Dietary intake of total energy, carbohydrate, protein, fat and dietary fiber, and moderate and hard physical activity were not different between the trials. 

### 3.2. Appetite

The VAS response curves for hunger, fullness, desire to eat, prospective food consumption, and thirst are presented in Figure 1 for both WM and LFC snacks. For hunger, WM produced lower ratings of hunger for up to 90-min post-consumption compared with baseline, while LFC significantly reduced hunger for only up to 20-min post-consumption (Figure 1A). Feelings of hunger were significantly greater for LFC compared with WM at 20, 40, 60, and 90 min (*p* < 0.05) (Figure 1a). Participants were significantly more full between baseline and 90-min for WM, and from baseline to 40-min for LFC (Figure 1b). Feelings of fullness were significantly greater for WM compared with LFC from 20-min to 90-min post-consumption (*p* < 0.05) (Figure 1b). Compared with baseline, desire to eat was significantly reduced up to 90-min post-WM consumption, while LFC only reduced the desire to eat for up to 20-min post-consumption (Figure 1c). Desire to eat was significantly greater for LFC compared with WM at all time points after consumption of snacks (*p* < 0.05) (Figure 1c). Similar to desire to eat, prospective food consumption was significantly reduced for up to 90-min post-WM consumption compared with baseline, while there were no differences for the LFC at any of the time points (Figure 1d). Prospective food consumption significantly differed between snacks up to 90-min post-consumption (*p* < 0.05) (Figure 1d). Thirst was significantly reduced at 20-min compared with baseline for both WM and LFC, while feelings of thirst were significantly reduced after WM consumption compared with LFC consumption at 20 and 40-mins (*p* < 0.05) (Figure 1e).

### 3.3. Glucose and Insulin

No significant differences in blood glucose concentrations were found between snacks at baseline or at 1-h postconsumption. Furthermore, there was no change in blood glucose at 1-h postconsumption compared with baseline for either snack (Figure 2a). Blood insulin levels significantly increased for both snacks 1-h postconsumption compared with baseline (*p* < 0.05), but no significant differences were found between snacks (Figure 2b). Following the four-week interventions, serum glucose levels were not significantly different between the WM and LFC interventions. There were also no significant changes in serum glucose between baseline (5.10 ± 1.36 mmol/L vs. 5.20 ± 1.20 mmol/L) and week four (5.59 ± 1.90 mmol/L vs. 5.10 ± 0.90 mmol/L) within each intervention. Similarly, serum insulin levels were not significantly different between the WM and LFC interventions at baseline (25.4 ± 2.1 mIU/L vs. 27.4 ± 1.8 mIU/L) or week four (27.5 ± 2.0 mIU/L vs. 26.8 ± 2.0 mIU/L) and did not change significantly from baseline to week four within each intervention.

### 3.4. Appetite-Regulating Hormones

The appetite-regulating hormones leptin, ghrelin, adiponectin, and CCK were measured in blood samples taken at the baseline fasted state and 1-h after snack consumption (Table 2). WM reduced leptin levels (*p* = 0.017), and LFC showed a trend to reduce leptin (*p* = 0.057). WM resulted in higher ghrelin (*p* = 0.004), and LFC tended to increase ghrelin levels (*p* = 0.086). There was a trend toward higher adiponectin concentration after WM consumption (*p* = 0.055).

### 3.5. C-Reactive Protein

Serum CRP as an indicator of chronic inflammation was not significantly different between the WM and LFC interventions at baseline (7.21 ± 3.37 mg/L vs. 7.14 ± 2.94 mg/L) or week four (7.07 ± 3.18 mg/L vs. 7.19 ± 3.36 mg/L) and did not differ significantly over time within each intervention.

### 3.6. Serum Lipids

Lipid profiles are shown below in Figure 3. WM consumption significantly lowered TG (*p* = 0.046). TC was significantly lower after four weeks of the WM intervention compared with LFC (*p* = 0.024). WM consumption significantly increased HDL-C (*p* = 0.046), while there was a trend toward lower HDL-C following LFC consumption (*p* = 0.066). LDL cholesterol decreased significantly after four weeks of WM consumption and increased significantly after four weeks of LFC consumption (*p* = 0.011).

### 3.7. Oxidative Stress and Antioxidant Capacity

After four weeks of WM treatment, there was a trend toward lower TBARS, an indicator of lipid peroxidation (*p* = 0.091). The LFC intervention showed a trend toward higher lipid peroxidation from baseline to week four (*p* = 0.092) (Figure 4a). Lipid peroxidation was lower at four week of WM intervention compared to four weeks of LFC intervention (*p* = 0.034). Levels of catalase, an antioxidant enzyme, were not significantly different between the two interventions. Total antioxidant capacity did not change significantly after four weeks of LFC consumption but increased significantly after four weeks of WM consumption (*p* = 0.003) (Figure 4b).

### 3.8. Liver Function Markers

There were no significant differences in the liver function markers AST, ALT, ALP, LDH, or CK after four weeks of the WM and LFC interventions. 

## 4. Discussion and Conclusions

As a natural food that provides fiber, micronutrients, and bioactive phytochemicals, watermelon may be a healthier alternative to conventional snacks. This study compared the effects of consuming watermelon versus an isocaloric low-fat cookie snack for four weeks on body weight, blood pressure, glucose and insulin concentrations, and biomarkers for inflammation, oxidative stress, and liver function. In addition, the acute effects of the two snacks were determined by measuring perceived appetite, blood glucose, insulin, and appetite-regulating hormone concentrations for up to 120 min post-consumption.

After four weeks, body weight and BMI increased in the LFC trial and decreased in the watermelon trial. Only a few intervention studies have compared a fruit snack with a processed snack, and none showed significant effects on body weight or other measures of adiposity [11,26,27]. Watermelon consumption has been associated with reduced weight gain and fat mass in some animal studies [18,28]. In humans, however, watermelon feeding studies with a duration of up to six weeks have not reported changes in body weight or body composition [21,22,29,30,31,32]. A possible explanation for these divergent results is that prior studies used watermelon juice or reconstituted watermelon powder, whereas our study used whole watermelon flesh. Whole fruit has been shown to promote greater satiety than juice [33]. This effect could be attributed to higher fiber content, which promotes satiety and reduces energy intake [34,35]. However, Flood-Obbagy and Rolls found that whole apples were more satiating than apple juice or applesauce, despite having similar fiber content [33]. Whole fruit has greater volume and requires more chewing than other forms, which could affect food intake by initiating cephalic-phase responses related to digestion and metabolism [33]. In addition, subject expectations that solid foods are more filling than juice may contribute to the satiating effects of whole fruit [33].

Acute watermelon consumption resulted in significantly greater satiety ratings than the LFC snack. The watermelon snack left subjects feeling significantly less hungry compared with baseline for up to two hours after snack consumption, while the LFC snack only reduced hunger for up to 20 min. Compared with the LFC snack, the watermelon snack had significantly greater volume due to its high water content. The volume of food can contribute to increased satiety through such factors as perception of the amount of food being consumed, time spent consuming food, rate of gastric emptying, and degree of gastric distension [36,37]. The sensory and hedonic characteristics of food such as sight, smell, texture, and taste can also influence palatability and satiety [38]. Appetite ratings reflect subjective feelings of satiety, and therefore concentrations of hunger and satiety hormones may provide better insight into the mechanisms controlling hunger and fullness. Watermelon consumption tended to increase adiponectin, which may partially explain the higher satiety effects of watermelon consumption. 

Because the watermelon snack contained almost twice as much total sugar (17 g) as the LFC snack (9 g), it would be expected to produce a higher postprandial glucose concentration. However, postprandial glucose and insulin were not significantly different between the two snacks. In previous studies, fruit produced smaller increases in blood glucose [8,39,40] and insulin [8,39,41] compared with processed foods. Although the watermelon snack contains sugar, the total load was quite low; therefore, the blood glucose concentrations apparently returned to normal within one hour. Additionally, watermelon possesses other nutritional components that may have suppressed a rise in blood glucose. Watermelon contains a small amount of dietary fiber, which can improve glucose tolerance [42]. What is likely of greater importance is that more than half of the total sugars in watermelon consist of fructose, which has little effect on blood glucose levels [12,39]. In fact, co-ingestion of fructose and glucose blunts the glycemic response to glucose, possibly by enhancing hepatic glucose uptake [39,43,44]. A meta-analysis showed that moderate intakes of up to 36 g/day of fructose (less than the 10.3 g in two cups of watermelon) resulted in lower fasting glucose and hemoglobin A1c (HbA1c) without causing elevated triglycerides [45]. Adiponectin levels were not significantly different between the trials, although there was a trend toward increased postprandial adiponectin following watermelon consumption. Adiponectin is a hormone secreted by adipose tissue that is paradoxically more abundant in obese than in lean individuals, and circulating levels increase after weight reduction [46,47]. As adiponectin increases insulin sensitivity, the trend toward increased postprandial adiponectin following watermelon consumption may suggest a glucose-stabilizing effect.

After four weeks, the watermelon intervention showed a trend toward lower SBP compared with baseline. In contrast, both SBP and DBP were higher after four weeks of LFC consumption. Several previous studies have shown greater decreases in SBP and DBP with watermelon supplementation compared with a refined carbohydrate control [22,29,30,31,48]. Watermelon is the richest natural source of l-citrulline, a nonessential amino acid that may be responsible for watermelon’s hypotensive effects [17]. l-citrulline is readily converted to l-arginine, and oral l-citrulline intake has been shown to raise circulating l-arginine levels [49]. In turn, endothelial nitric oxide synthase (eNOS) converts l-arginine to nitric oxide, which induces vascular smooth muscle relaxation [17].

Four weeks of watermelon consumption resulted in an improved blood lipid profile, including lower levels of TG and LDL-C and higher levels of HDL-C. Following the LFC intervention, TG and LDL-C increased and HDL-C trended lower. One prior human study showed that six weeks of supplementation with watermelon extract reduced TC and LDL-C compared with a carbohydrate beverage [21]. Several animal studies have also shown beneficial effects of watermelon consumption on TC, LDL-C, and TG levels [18,19,20]. Analysis of hepatic mRNA expression in rats found that watermelon reduced expression of fatty acid synthase (FAS), an enzyme involved in fatty acid synthesis, and HMG-CoA reductase (HMGCR), the rate-limiting enzyme in cholesterol synthesis [20]. These metabolic alterations may have contributed to the improved lipid profile.

Watermelon consumption for four weeks increased antioxidant status and tended to reduce oxidative stress. One previous human study reported increased antioxidant status as indicated by higher ferric reducing ability of plasma (FRAP) and oxygen radical absorbance capacity (ORAC) following two weeks of watermelon supplementation [32]. Watermelon consumption has also been associated with lower oxidative stress and increased antioxidant status in experimental animals [18,19,20]. These results may be partially explained by increased plasma concentrations of lycopene and other carotenoid antioxidants [50,51]. Watermelon’s l-citrulline content may also reduce oxidative stress by serving as a substrate for endogenous nitric oxide production [17]. Although nitric oxide is primarily known for its vasodilatory properties, it can also reduce oxidative stress by scavenging or preventing the formation of hydroxyl radicals [52].

An important limitation of this study is the measurement of postprandial response at a single 60-min time point. The current study found no significant differences in postprandial glucose or insulin levels between trials. The protocol may have not been able to detect genuine differences if concentrations of these biomarkers peaked before the second blood draw. In support of this hypothesis, some studies have found glucose to peak approximately 30 min after fruit consumption and decline thereafter [8,39]. For example, in the study by Furchner-Evanson et al., plasma glucose concentrations did not differ significantly between the dried plum and cookie trials at the 60-min timepoint [8]. However, due to the higher values at the 15-, 30-, and 45-min timepoints, the cookie trial produced a significantly higher glucose level that would not have been detected if blood would have been collected only one hour after consumption. Future studies should include more critical time points for measuring satiety hormones—i.e., measurements at the 15-, 30-, and 45-min marks to define postprandial peak. Because the four-week WM trial preceded the LFC trial for all subjects, it is possible that there was an order effect. Subjects were told that the purpose of the study was to compare effects of different snacks on health, but they did not know that the specific focus was on watermelon versus cookies. This aspect of the study design may have reduced the risk of the placebo or order effect. A nested design, in which half the subjects start with watermelon and the other half with LFC, should be used to reduce the chance of order effects in the future study. Another limitation is that it is unclear which biochemical components or sensory qualities of watermelon contributed to increased satiety and reduced body weight compared with the LFC control. In order to further investigate these effects, it would be necessary to test whole watermelon against its isolated bioactive components. Finally, it is unknown whether the watermelon dosage was optimal, or whether a larger dosage might have had more significant effects on measured outcomes. Therefore, future studies should compare the effects of different watermelon dosages on satiety and body weight.

In conclusion, watermelon promoted greater satiety than an isocaloric LFC snack for up to 90 min post-consumption. Additionally, four weeks of watermelon consumption reduced body weight and blood pressure while improving blood lipid profile and antioxidant status. These results suggest that fresh watermelon, when consumed in place of conventional refined carbohydrate snacks, may help reduce appetite and assist with weight management while reducing cardiovascular risk factors.

## Figures and Tables

**Figure 1 nutrients-11-00595-f001:**
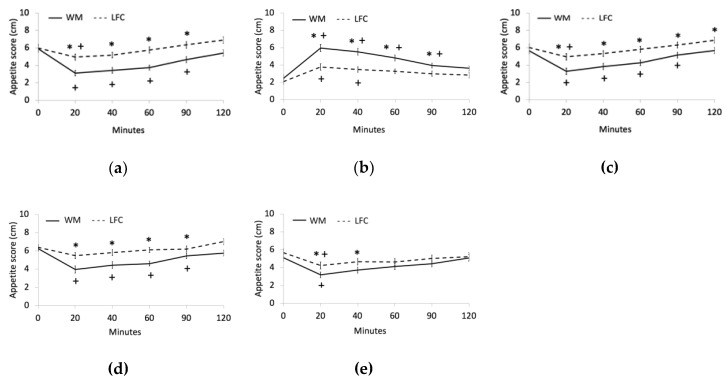
Effects of watermelon (WM) and low-fat cookies (LFC) on (**a**) hunger, (**b**) fullness, (**c**) desire to eat, (**d**) prospective food consumption, and (**e**) thirst. *: different between WM and LFC; +: different from baseline. VAS: visual analogue scale.

**Figure 2 nutrients-11-00595-f002:**
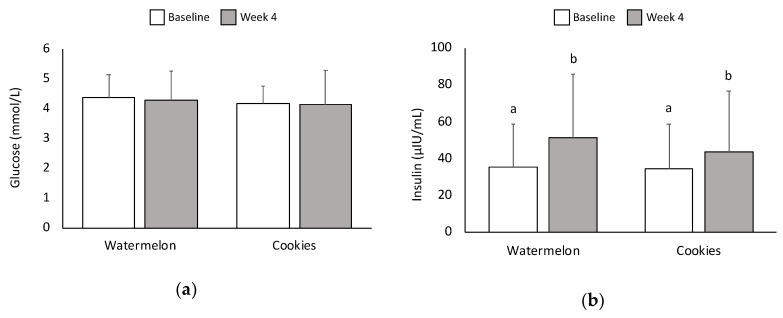
(**a**) Effects of WM and LFC on postprandial glucose. No significant differences in blood glucose were observed between snacks and between pre-consumption (pre) and 1-h postconsumption (post). (**b**) Effects of WM and LFC on postprandial insulin. Blood insulin significantly increased (*p* < 0.05) in both snacks 1-h postconsumption compared with baseline. Data are presented as means ± SD. Within a variable, values not sharing common superscript are significantly different at *p* < 0.05.

**Figure 3 nutrients-11-00595-f003:**
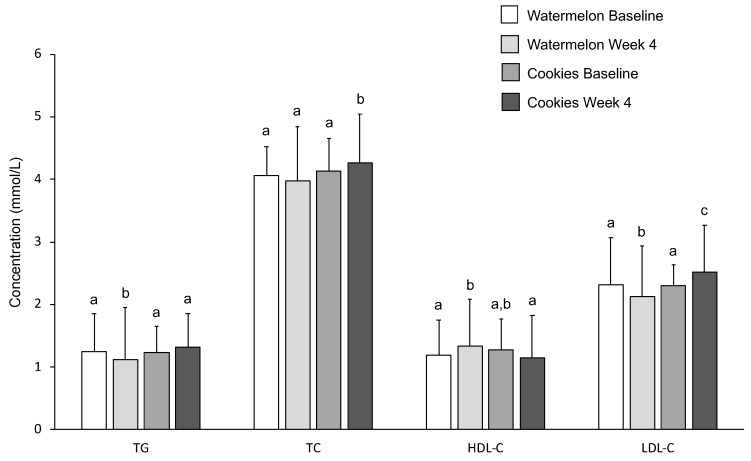
Effects of WM and LFC on serum concentrations of triglycerides (TG), total cholesterol (TC), high-density lipoprotein cholesterol (HDL-C), and low-density lipoprotein cholesterol (LDL-C) at baseline and week 4 of each intervention. Data are presented as means ± SD. Within a variable, values not sharing a common superscript are significantly different at *p* < 0.05.

**Figure 4 nutrients-11-00595-f004:**
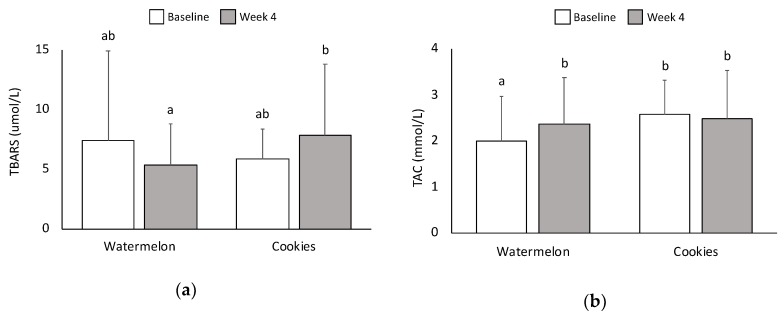
Effects of WM and LFC on serum values for (**a**) thiobarbituric acid reactive substances (TBARS) and (**b**) total antioxidant capacity (TAC) at baseline and week 4 of each intervention. Data are presented as means ± SD. Within a variable, values not sharing a common superscript are significantly different at *p* < 0.05.

**Table 1 nutrients-11-00595-t001:** Effects of snacks on body weight, body mass index (BMI), blood pressure, waist-to-hip ratio, and body fat of participants at baseline and week 4 for each intervention.

Measurements	Watermelon (*n* = 33)		Cookies (*n* = 33)	
	Baseline	Week 4	Baseline	Week 4
Body Weight (kg)	89.4 ± 15 ^a^	88.9 ± 16 ^b^	89.3 ± 16 ^a^	89.9 ± 16 ^c^
BMI	30.5 ± 3.5 ^a^	30.4 ± 3.7 ^b^	30.5 ± 3.7 ^a^	30.7 ± 3.8 ^c^
SBP (mm Hg)	127 ± 15 ^a,b^	125 ± 14 ^a^	124 ± 14 ^a^	129 ± 14 ^b^
DBP (mm Hg)	79.9 ± 7.2 ^a^	79.6 ± 9.7 ^a^	77.2 ± 9.0^b^	81.2 ± 10 ^a^
W/H ratio	0.850 ± 0.06 ^a,b^	0.845 ± 0.07 ^a^	0.847 ± 0.07 ^a,b^	0.857 ± 0.07 ^b^
Body Fat (%)	37.8 ± 8.2 ^a^	38.0 ± 8.5 ^a^	37.7 ± 8.2 ^a^	38.2 ± 7.9 ^b^

Data are presented as means ± SD. Data within rows with varying superscript letters are statistically different (*p* < 0.05). *n* = 33 (20 males/13 females). BMI, body mass index; SBP, systolic blood pressure; DBP, diastolic blood pressure; W/H ratio, waist-to-hip ratio.

**Table 2 nutrients-11-00595-t002:** Effects of WM or LFC consumption on postprandial satiety hormone.

	Watermelon (*n* = 33)		Cookies (*n* = 33)	
	Pre	Post	Pre	Post
Leptin (ng/mL)	3.65 ± 2.02 ^a^	3.28 ± 2.01 ^b^	3.71 ± 1.96 ^a^	3.42 ± 2.07 ^a,b^
Ghrelin (pg/mL)	414 ± 246 ^a^	520 ± 356 ^b^	424 ± 250 ^a^	511 ± 352 ^a,b^
Adiponectin (μg/mL)	9.93 ± 5.81	10.66 ± 5.08	9.20 ± 5.63	8.41 ± 6.33
CCK (pg/mL)	465 ± 242	514 ± 170	495 ± 178	542 ± 207

Values are means ± SD. *n* = 33 (20 males/13 females). Data within rows with varying superscript letters are statistically different *p* < 0.05.

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
