# Peer review of "Effects of Fresh Watermelon Consumption on the Acute Satiety Response and Cardiometabolic Risk Factors in Overweight and Obese Adults"

_nutrients, 2019, doi:10.3390/nu11030595_

Reviewer 1 Report

The study “Effects of fresh watermelon consumption on the acute satiety response and cardiometabolic risk factors in overweight and obese adults” aimed to determine the effect of watermelon consumption compared to low fat cookies as snack in overweight to healthy individuals. Overall, the study in appropriately designed and the conclusions well supported by the data. Despite a relatively low effect size, the publication is of good quality and well written but the authors should address the following point to be suitable for publication.

Major Comments:

1.       In Table 1, the authors should present the result of the ANOVA for the different parameters (i.e. Watermelon vs LFC effect, Intervention effect (week 4 vs baseline) and interaction factor). Given that the statistics applied should be paired, SD would be more relevant than SE.   

2.       The authors should acknowledge in the discussion that, despite being technically impossible, the participants were not blinded, thus, given the perception that eating fruits is healthy a placebo effect cannot be entirely ruled out.  

3.       It would be informative to provide the  qukitative and quantitative composition in carbohydrates, lipids, proteins, fibers, etc of the WM and LFC.

4.       Line 230, the insulin levels in the text do match the ones of the graph (25-30 versus 30-50).

Minor Comments:

1.       Line 186: The authors should write that waist to hip ratios tends to be lowered, since the p-value does not reach the significant threshold.

2.       The authors should also use I.S values for their measurements (e.g. glycaemia in mM, etc).

Author Response

Reviewer 1

Comments and Suggestions for Authors

The study “Effects of fresh watermelon consumption on the acute satiety response and cardiometabolic risk factors in overweight and obese adults” aimed to determine the effect of watermelon consumption compared to low fat cookies as snack in overweight to healthy individuals. Overall, the study in appropriately designed and the conclusions well supported by the data. Despite a relatively low effect size, the publication is of good quality and well written but the authors should address the following point to be suitable for publication.

Major Comments:

1. In Table 1, the authors should present the result of the ANOVA for the different parameters (i.e. Watermelon vs LFC effect, Intervention effect (week 4 vs baseline) and interaction factor). Given that the statistics applied should be paired, SD would be more relevant than SE.

Response: In Table 1, some results are based on nonparametric (Wilcoxon) analysis. For these results, there are no p-values for main effects or interaction, so we did not include these values in the table. However, when applicable, we have added p-values for main effects or interactions in the text. SE was replaced with SD.

2. The authors should acknowledge in the discussion that, despite being technically impossible, the participants were not blinded, thus, given the perception that eating fruits is healthy a placebo effect cannot be entirely ruled out.

Response: Thank you for your comment. One consideration is that subjects in our study were told that the purpose of the study was to compare the effects of different snacks on health, but they did not know that the specific focus was on watermelon versus cookies. This aspect of the study design may reduce the risk of the placebo effect you describe. This has been addressed in discussion (LL 384-388).

3. It would be informative to provide the qualitative and quantitative composition in carbohydrates, lipids, proteins, fibers, etc of the WM and LFC.

Response: The requested information has been added to the methods section (LL103-105).

4. Line 230, the insulin levels in the text do match the ones of the graph (25-30 versus 30-50).

Response: We believe that the data and graph are correct. Figure 2b shows preprandial and postprandial insulin values, whereas the data in lines 238-239 refers to values before and after 4 weeks of treatment.

Minor Comments:

1. Line 186: The authors should write that waist to hip ratios tends to be lowered, since the p-value does not reach the significant threshold.

Response: Waist-to-hip data was analyzed using the Wilcoxon non-parametric test, thus the p-value is significant at 0.019. The manuscript has been updated with the correct p-value.

2. The authors should also use I.S. values for their measurements (e.g. glycaemia in mM, etc).

Response: Figure 2a has been changed to show the vertical axis in mmol/L.

Thank you.

Reviewer 2 Report

This is a very interesting paper comparing the effects of watermelon to low fat cookie consumption on satiety, postprandial glucose and insulin response, and adiposity and body weight change after 4 weeks of intervention in overweight and obese adults. The paper is structured, succint and well-written. The strengths and limitations of the study are well formulated. I have a few questions on the study design and statistical analysis issues. In methods section it should be mentioned how the sample size was derived at using a crossover randomized trial and whether it was powered to detect significant interactions. Also, what is the trial registration number? This can be added just after the abstract. Regarding the statistical analysis performed, the authors didn't state whether the normality assumption was tested for each continuous outcome, and if it was ever violated which statistical model and test were used; eg, did they use nonparametric ANOVA or transformed the outcome variable? Also, for pairwise comparisons which test was used , eg, sign test ? This should be clarified in methods section. A major flaw for posthoc comparisons is not to perform any correction for multiplicity. The authors didn't state anything in regards to this, so I am assuming that it was not performed. How were the ANOVA models adjusted? For example, for analysing a crossover trial, each ANOVA model should control for subjects nested within the sequence. This should be explicity stated in methods and below the table representing the ANOVA results. I suggest that the authors resubmit the paper with the abovementioned changes.

The reviewer        

Author Response

Reviewer 2

Comments and Suggestions for Authors

This is a very interesting paper comparing the effects of watermelon to low fat cookie consumption on satiety, postprandial glucose and insulin response, and adiposity and body weight change after 4 weeks of intervention in overweight and obese adults. The paper is structured, succint and well-written. The strengths and limitations of the study are well formulated. I have a few questions on the study design and statistical analysis issues.

Comment: In methods section it should be mentioned how the sample size was derived at using a crossover randomized trial and whether it was powered to detect significant interactions.

Response: Sample size was determined based on a previous human trial of watermelon (reference #22). Based on that study, power analysis (G*Power, Germany) indicated that significant differences in blood pressure could be obtained in a sample of 33 subjects at 75% power and an alpha-level of P<0.05. This information has been added to the methods section (LL82-84).

Comment: Also, what is the trial registration number? This can be added just after the abstract.

Response: The trial registration number is NCT03380221 and it has been added in the abstract.

Comment: Regarding the statistical analysis performed, the authors didn't state whether the normality assumption was tested for each continuous outcome, and if it was ever violated which statistical model and test were used; eg, did they use nonparametric ANOVA or transformed the outcome variable? Also, for pairwise comparisons which test was used, eg, sign test ? This should be clarified in methods section. A major flaw for posthoc comparisons is not to perform any correction for multiplicity. The authors didn't state anything in regards to this, so I am assuming that it was not performed. How were the ANOVA models adjusted? For example, for analysing a crossover trial, each ANOVA model should control for subjects nested within the sequence. This should be explicity stated in methods and below the table representing the ANOVA results. I suggest that the authors resubmit the paper with the abovementioned changes.

Response: The normality of the data was checked using the explore function in SPSS. When data showed a normal distribution, a two-way repeated measures analysis was used to analyze the effects of WM and LFC treatments on each of the variables over time, as well as any significant interactions between snack type and time. If the interaction was significant, a LSD analysis was used for post-hoc comparison. When the normality assumption was suspect, a nonparametric (Wilcoxon) analysis was performed. The statistical analysis section of the methods has been revised with these details (LL179-184).

You are correct that nesting within the sequence is preferable as a study design in order to control for order effects. Due to the seasonality of watermelon, all subjects were given watermelon first followed by a 2-4 week washout period, and then crossed over to an isocaloric-matched low-fat cookie snack. Due to this study design, there may have been an order effect. This limitation and future direction have been added to the discussion section (LL383-388). By the way, one consideration is that subjects in our study were told that the purpose of the study was to the compare effects of different snacks on health, but they did not know that the specific focus was on watermelon versus cookies. This aspect of the study design may reduce the risk of the placebo or order effect.

Thank you.